# DELTA: Integrating Multimodal Sensing with Micromobility for Enhanced Sidewalk and Pedestrian Route Understanding

**DOI:** 10.3390/s24123863

**Published:** 2024-06-14

**Authors:** Alireza Akhavi Zadegan, Damien Vivet, Amnir Hadachi

**Affiliations:** 1ITS Lab, Institute of Computer Science, University of Tartu, 51009 Tartu, Estonia; amnir.hadachi@ut.ee; 2ISAE-SUPAERO, Universite de Toulouse, 31400 Toulouse, France; damien.vivet@isae-supaero.fr

**Keywords:** urban analytics, pedestrian area mapping, multimodal dataset

## Abstract

Urban environments are undergoing significant transformations, with pedestrian areas emerging as complex hubs of diverse mobility modes. This shift demands a more nuanced approach to urban planning and navigation technologies, highlighting the limitations of traditional, road-centric datasets in capturing the detailed dynamics of pedestrian spaces. In response, we introduce the DELTA dataset, designed to improve the analysis and mapping of pedestrian zones, thereby filling the critical need for sidewalk-centric multimodal datasets. The DELTA dataset was collected in a single urban setting using a custom-designed modular multi-sensing e-scooter platform encompassing high-resolution and synchronized audio, visual, LiDAR, and GNSS/IMU data. This assembly provides a detailed, contextually varied view of urban pedestrian environments. We developed three distinct pedestrian route segmentation models for various sensors—the 4K camera, stereocamera, and LiDAR—each optimized to capitalize on the unique strengths and characteristics of the respective sensor. These models have demonstrated strong performance, with Mean Intersection over Union (IoU) values of 0.84 for the reflectivity channel, 0.96 for the 4K camera, and 0.92 for the stereocamera, underscoring their effectiveness in ensuring precise pedestrian route identification across different resolutions and sensor types. Further, we explored audio event-based classification to connect unique soundscapes with specific geolocations, enriching the spatial understanding of urban environments by associating distinctive auditory signatures with their precise geographical origins. We also discuss potential use cases for the DELTA dataset and the limitations and future possibilities of our research, aiming to expand our understanding of pedestrian environments.

## 1. Introduction

The urban landscape is undergoing a significant transformation influenced by the increasing integration of mobile vehicles, last-mile delivery services, micro-mobility solutions, and advanced technologies poised to reshape urban mobility. This evolution highlights the critical role of pedestrian infrastructure, such as sidewalks, crosswalks, pedestrian routes, and public pathways. These elements are not only pathways but essential components of the contemporary urban environment, where human mobility seamlessly interacts with emerging autonomous and mobile robotic systems [1]. Additionally, the diversity of cities across regions significantly shapes our mobility models. Urban design and infrastructure are not just influenced by technological advancements but also by regional characteristics that dictate the functional and cultural aspects of mobility, supporting the view that cities mold our mobility paradigms, as further discussed in recent studies [2,3].

Urban transportation and planning are intricately linked through a dynamic feedback loop, where transportation infrastructure not only shapes urban form but also influences population density, land-use patterns, and economic activities. Conversely, city planning decisions dictate the demand and feasibility of various transportation systems [4]. This interplay is further impacted by technological advancements [5]. For instance, the rise of automobiles in the 20th century led to urban sprawl and car-dependent developments, while today’s advancements in autonomous vehicles, shared mobility options, and electric cars hold the potential to reshape urban landscapes once again.

As we transition into a new era of urban living, one truth emerges with clarity: these commonly overlooked urban spaces, primarily designed for pedestrian use, are no longer exclusive domains of human foot traffic. Instead, they are becoming dynamic environments where diverse modes of mobility coexist. Electric scooters, e-bikes, autonomous delivery robots, and future technologies like autonomous personal mobility devices are increasingly becoming part of the urban fabric [6]. They share the same physical spaces and temporal constraints as pedestrians. Thus, a fundamental challenge arises, one that underscores the necessity for a comprehensive understanding of pedestrian infrastructure in this shifting urban paradigm. These transformations raise questions about safety, efficiency, and accessibility in shared urban spaces, necessitating a nuanced and data-driven approach to urban planning and technology development.

To tackle the intricate challenges facing urban planning and navigation technologies within the rapidly evolving urban landscapes, we introduce the DELTA dataset. This dataset is a comprehensive response to the multifaceted dynamics of urban pedestrian environments, designed to facilitate advanced research and development in this domain. A cornerstone of our research is the deployment of a custom-designed modular multi-sensing e-scooter platform, which captures all data simultaneously in a single pass. This platform integrates a diverse array of sensors, including a high-resolution 4K monocular camera, stereo camera, LiDAR, audio recording devices, IMU, and GNSS receiver. Each sensor captures unique environmental aspects with varying degrees of resolution and data types, all synchronized using a unified time-stamping system. This approach ensures a rich, multidimensional dataset, capturing the complexity of urban environments effectively and efficiently.

The DELTA dataset includes sequences of stereo RGB images, depth maps, and point clouds generated from disparity maps for detailed visual representations, alongside sequences of 3D point clouds, reflectivity, and range channel data from LiDAR for precise spatial measurements. Additionally, it features raw GNSS data and IMU measurements for accurate positioning and motion tracking. To effectively interpret the nuanced data collected from this varied sensor array, we have developed three distinct sidewalk and pedestrian route segmentation models for the 4K images, stereo camera RGB images, and the LiDAR’s reflectivity channel, each tailored to the specific characteristics of the sensor it is designed for.

Moreover, the dataset is enriched with two sources of raw audio files, which are synchronized with the visual data to provide a comprehensive multisensory perspective of the urban environment. An event-based classification model further processes these audio files, identifying and classifying urban sounds. These classified audio files, segmented at 5 s intervals and geotagged, offer an immersive auditory layer to the dataset, deepening our understanding of urban soundscapes. Figure 1 presents an overview of the platform, showcasing our multisensor datasets, segmentation techniques, sound event classifications, and the area under study.

Our paper is structured as follows. In Section 2, we review existing urban datasets and highlight the need for more comprehensive pedestrian-focused data, demonstrating the limitations of current road-centric resources. Section 3 details our innovative multi-sensor platform used for data collection, emphasizing the integration and synchronization of various data types like visual, LiDAR, and audio. In Section 4, we detail the methodologies employed to ensure precise alignment, synchronization, and cleaning of the collected data. In Section 5, we discuss the role of auditory analysis in urban studies and the classification of urban sounds and their importance in understanding the urban acoustic environment. In Section 6, we present our approach to segmenting pedestrian pathways. The potential applications of our dataset are explored in Section 7. In Section 8, we reflect on the challenges encountered and outline future research directions. Section 9 provides a summary of our findings and the implications for future urban studies.

## 2. Related Works

The collection and availability of high-quality datasets play a pivotal role in advancing research and development in fields like computer vision, autonomous navigation, and urban planning. Although many datasets have been developed for urban research, there is a clear lack of data on important but often ignored parts of the city like sidewalks and pedestrian routes. This section provides an overview of existing datasets and highlights the unique contribution of our work in addressing this gap.

**Road-Centric Datasets:** Many existing datasets predominantly focus on data collected from vehicular perspectives and road networks, reflecting the significant emphasis on mobile vehicles in the realm of autonomous navigation and traffic management. Notable examples include the KITTI vision benchmark suite [7], KITTI-360 [8], Cityscapes [9], Oxford [10], ApolloScape [11], and Waymo [12], while these resources have been instrumental in advancing research in autonomous driving and urban traffic systems, they are primarily oriented towards vehicular environments. Similarly, while datasets like UrbanLoco [13] and Hong Kong UrbanNav [14] attempt to fill the gap by including raw GNSS measurements for detailed navigation research, they mainly focus on highly urbanized, vehicle-centric environments. These datasets, with their comprehensive coverage, have set the standard for autonomous vehicle technologies and traffic analysis methodologies. However, their predominant focus on vehicular viewpoints often leads to the neglect of the intricacies and unique challenges present in pedestrian areas. This oversight in existing datasets underscores a critical need in the field for resources that provide detailed insights into pedestrian spaces, essential for a more holistic understanding of urban mobility and safety. Such comprehensive data would enable the development of autonomous systems capable of navigating and operating safely in the full spectrum of urban environments, addressing both vehicular and pedestrian dynamics.

**Sidewalk-Centric Datasets:** While a segment of existing datasets provides insights into pedestrian dynamics, they primarily capture limited scenarios such as crowd behavior in public spaces or indoor tracking. These specialized datasets, although valuable, do not reflect the full spectrum of pedestrian interactions within the diverse urban landscapes encountered daily. Recent advances in deep learning have notably enhanced computer vision techniques like semantic segmentation, offering more granular analysis of urban scenes. Despite this, the field still grapples with a significant shortage of quality training data for pedestrian infrastructure analysis. In recent research endeavors within urban data collection for pedestrian infrastructure, a variety of innovative methodologies have been utilized to develop specialized datasets. Among these, Tile2Net [15] employs semantic segmentation of aerial imagery to map pedestrian infrastructures, showcasing an aerial perspective in data collection. SideGuide [16] takes a unique approach by integrating feedback from interviews with impaired individuals, focusing on sidewalk-specific objects and combining crowd-sourced and stereo data to form its dataset. The study by [17] represents a distinct method, using Google Street View (GSV) panoramas for assessing sidewalk accessibility. This approach utilizes existing online map imagery as a primary data source. In contrast, the PESID [18] project concentrates on pedestrian safety, developing its dataset from a collection of labeled images depicting various sidewalk environments. Further diversifying the data collection techniques, ref. [19] combined results from both aerial and street view imagery to create a comprehensive sidewalk network dataset, this integrative method synthesizes data from multiple sources. CitySurfaces [20] applies computer vision techniques to classify sidewalk materials using street-level images, offering a dataset focused on material classification. An intriguing methodology was presented by ref. [21], where the 3D virtual environment of the CARLA simulator [22] was utilized to develop a dataset specifically for Electric Wheelchairs (EW) on sidewalks. This approach illustrates the innovative use of synthetic scenes, showcasing their growing relevance in urban data collection. These diverse methodologies highlight the evolving landscape of urban data collection for pedestrian infrastructure. Each approach, whether it leverages advanced image processing, virtual environments, or crowd-sourced data, contributes uniquely to the field, enriching our tools and understanding for urban planning and pedestrian infrastructure development.

## 3. The DELTA Platform 

In this section, we introduce the DELTA platform (depicted in Figure 2) which is specifically engineered for collecting a multimodal dataset targeting urban areas often neglected in typical urban studies. The platform is based on a modified electric scooter equipped with a comprehensive array of advanced sensors, enabling it to capture a wide range of environmental data from pedestrian pathways and sidewalks—spaces that are typically underrepresented in data collections.

At the heart of this system is a high-precision GNSS/IMU unit, essential for accurate location tracking and motion data, which are critical for precise mapping and effective data integration. For visual and spatial data acquisition, the platform includes a ZED2i stereocamera and a 360-degree Ouster LiDAR, allowing for an extensive and detailed capture of environmental features. This is complemented by a 4K monocular camera and a stereo audio recorder, which augment the dataset with rich visual and auditory data. The specifications of the platform’s sensors are detailed in Table 1. Data processing is efficiently handled by an NVIDIA AGX Orin module serving as the central processing unit. The system is mounted on multiple damping and vibration isolation systems to minimize noise and vibrations as much as possible to maintain the integrity of the data collected. Furthermore, its modular design also allows for easy customization to meet diverse research requirements, significantly broadening the platform’s application in various urban studies.

Integrating these high-precision sensors onto a singular mobile platform significantly enhances its utility in urban data analytics, particularly in the realms of mobile robotics and micro-mobility services. This setup is pivotal for analyzing pedestrian traffic patterns, supporting the development and upkeep of urban infrastructure, and optimizing the efficiency and safety of mobile robotics and micro-mobility solutions in complex city environments.

### 3.1. LiDAR

A total of 15,497 scans were collected using Ouster OS1-64 LiDAR sensor operating at a 905 nm wavelength. This sensor boasts a 360° field of view (FOV), enabling comprehensive environmental coverage for our study. With a maximum range of 200 m and ±2 cm accuracy at 100 m, the Ouster LiDAR provided detailed capture of even distant objects. Each scan contained a 3D point cloud with 32-bit reflectivity data, and the range (depth) channel. Additionally, the LiDAR’s 10 Hz sampling frequency ensured a consistent data acquisition rate. Figure 3 shows the point cloud, range and reflectivity data extracted from the LiDAR.

### 3.2. Stereocamera

A total of 91,411 stereo image sequences were captured by the ZED2i stereocamera, a high-performance camera equipped with dual 1/3″ 4MP rolling shutter CMOS sensors. This camera boasts a wide FOV of 110° (horizontal) by 70° (vertical), allowing for comprehensive scene capture. The ZED2i offers a versatile depth sensing range, capturing objects from a minimum of 0.3 m to a maximum of 20 m. Each has a resolution of 720 × 405 pixels and was operated at 58 FPS. Utilizing the ZED camera API, we have extracted depth map and point cloud data from each stereo image pairs. These, along with the RGB images, comprise the three types of data formats present in our stereocamera dataset (refer to Figure 4).

### 3.3. 4K Camera

A total of 45,957 high-resolution images (4096 × 3072 pixels) were captured using an Osmo 3 action camera. This camera features a 1/1.7″ 12MP CMOS sensor and a wide field of view of 155°. The high resolution aided in capturing detailed information about the urban environment. The camera operated at a frame rate of 30 FPS, suitable for capturing dynamic scenes and ensuring smooth transitions in our visual data collection.

### 3.4. GNSS

The dataset encompasses comprehensive Global Navigation Satellite System (GNSS) measurements from multiple satellite constellations, namely GPS, GLONASS, Galileo, and BeiDou. These measurements are essential for high-precision navigation and geospatial analysis, providing an accuracy of approximately 1.1 m. The dataset comprises three distinct data types, pseudorange, carrier phase, and Doppler shift, each offering different levels of insight into distance and velocity measurements, crucial for applications that require enhanced accuracy, such as differential GNSS. Each satellite data point within the dataset is characterized by unique identifiers (svId), signal quality metrics (carrier-to-noise ratio (CnO), lock time), and detailed navigation outputs (UBX-NAV-PVT message). These navigation outputs report position, velocity, and time information, rendering the dataset valuable for real-time applications and urban planning endeavors. Furthermore, the dataset facilitates post-processed kinematics (PPK), enabling centimeter-level accuracy improvements. Figure 5 shows the spatial coverage and the trajectory of our data collection.

### 3.5. Inertial Measurement Unit (IMU)

The dataset includes high-resolution inertial and environmental parameters such as tri-axial acceleration, gyroscope readings, angular orientation, magnetometer readings, and ambient temperature, all timestamped for precise data logging. Integrating this IMU data with GNSS and GIS is essential for advanced navigation systems used in mobile robotics and micro-mobility services [23]. GNSS provides accurate positional data, while GIS offers digital maps with essential spatial environmental information.

Gyroscopes in the IMU ensure the accurate orientation of maps, critical for safe navigation. Magnetometers offer crucial directional information, particularly useful in areas with complex magnetic fields that might disrupt traditional compass readings [24]. Additionally, synchronizing IMU data with GNSS coordinates enhances the precision of the location data used for creating maps within GIS systems. The use of advanced data fusion techniques such as the Kalman filter [25], extended Kalman Filter [26], and fuzzy logic systems [27] in conjunction with accelerometer data significantly improves the mapping system’s functionality and accuracy. These techniques allow for a more reliable understanding of the robot’s position and orientation, as captured by GIS. By leveraging GNSS, GIS, and IMU data with these sensor fusion techniques, navigation systems can achieve high levels of accuracy and robustness, enabling the safe and efficient operation of mobile robots and micro-mobility services in complex urban settings.

### 3.6. Audio

The Tascam DR-07X audio recorder and the Osmo 3 action camera were utilized to capture 25 min of raw audio, each selected for their unique capabilities. The Tascam DR-07X was chosen for its superior audio quality, featuring a high-fidelity cardioid condenser stereo pair and a sampling frequency of 96 kHz/24-Bit, ideal for recording detailed soundscapes. In parallel, the Osmo 3, equipped with two dedicated microphones and a camera, was used to record ambient urban sounds while also enabling the seamless integration of audio with visual data.

## 4. Data Cleaning and Synchronization 

Data cleaning is a crucial stage in preparing our dataset for robust analyses and application in urban mobility and planning studies. Given the complexity of urban environments and the variety of sensors involved, systematic cleaning procedures were essential to ensure the consistency, accuracy, and reliability of our multimodal dataset. After completing our data collection routes, we sampled GPS data at 4 m intervals. This strategy was selected to strike a balance between comprehensiveness and manageability, ensuring an accurate representation of the traveled paths without redundancy. The chosen interval resulted in the collection of 950 distinct GPS points across the surveyed area, a figure that effectively represents the total length of the routes divided by the sampling interval. For each of these 950 GPS points, timestamps were recorded, capturing the exact moment each point was logged. These timestamps are vital for synchronizing the GPS data with sensor data from a variety of sources including GPS, IMU, stereocamera, and LiDAR, which capture complementary information at varying frequencies.

To synchronize the GPS data with the data collected from different sensors, we employed a two-pronged strategy to achieve effective synchronization:**Automated Synchronization with Jetson Clock and KD-Tree:** We employ the Jetson’s internal system clock as the primary reference for all sensor timestamps to minimize potential discrepancies due to clock drift from individual sensors. Each sensor reading is timestamped at the exact moment of capture, referenced to the Jetson clock, ensuring consistency across the dataset. Following data acquisition, we utilize the KD-tree algorithm [28] for post-processing alignment. This algorithm efficiently organizes timestamps into a k-dimensional space, allowing for rapid nearest-neighbor searches. By employing this method, we are able to accurately match each GPS point with the closest corresponding frames from the stereocamera and each LiDAR scan based on temporal proximity. This synchronization process ensures that the GPS data aligns perfectly with the data collected from different sensors, thus maintaining the integrity and accuracy of our multimodal dataset.**Manual Synchronization for Image Data:** For the manual synchronization phase, we developed an intuitive graphical user interface (GUI) that facilitates seamless navigation through two key datasets: RGB images from the stereocamera, which are synchronized with the Jetson system clock and serve as a temporal reference, and the extracted 4K images from the Osmo 3 video footage, which lack direct timestamps tied to the Jetson system clock. By manually browsing and comparing images from both datasets, we identified pairs that best matched and depicted the same scene at the exact moment. We then assigned the same timestamp to the manually selected high-resolution images as their counterpart in the stereo camera RGB dataset. Although this manual process is more time-consuming than automated synchronization, it is essential for ensuring the accurate temporal alignment of the high-resolution Osmo 3 images with data from other sensors. Figure 6 illustrates the GUI in action.

### Audio Synchronization

To synchronize audio recordings from the action camera with those from a Tascam-DR-07X audio recorder, we start by standardizing the sample rates of both sources, ensuring uniformity for accurate processing. The heart of this synchronization process is the use of cross-correlation, a statistical technique that measures the similarity between two audio waveforms. This involves determining the time lag, represented as τ, that maximizes the similarity between the two signals. The cross-correlation function, symbolized as R(τ) and defined in Equation (Equation 1), is instrumental in identifying the point of highest correlation between the two audio streams. This peak correlation signifies the optimal alignment point, indicating the exact time shift needed to achieve synchronization between the recordings. This method provides a precise mechanism for aligning audio files from different sources, ensuring their temporal coherence for further analysis or integration.
(1)$$R\left(\tau \right)=\sum_{n}x\left[n\right]\cdot y\left[n+\tau \right]$$

To achieve alignment, one of the audio signals needs to be adjusted in time by inserting a silent interval that matches the absolute value of the identified lag. This adjustment ensures that the inherent qualities of the audio files remain intact, even after incorporating the necessary time shift. By adding this silent period, we align the start times of the audio tracks from the action camera and the Tascam recorder, ensuring they begin simultaneously and are precisely synchronized. Figure 7 illustrates a segment where the two audio files have been successfully synchronized, demonstrating the effectiveness of this adjustment in achieving temporal coherence between different audio sources.

## 5. Audio Event-Based Classification

In environmental assessments, auditory analysis is a relatively new field compared to visual methods. Historically, urban sounds were often regarded as nuisances, primarily studied in the context of noise pollution [29]. Significant research has focused on developing outdoor noise measurement and reduction techniques [30]. The detrimental effects of noise from vehicles and industrial machinery on physical and mental well-being are well documented [31]. Traditionally, sound assessments relied on sound level meters, simplifying complex urban soundscapes into mere decibel measurements.

Recent years, however, have seen a paradigm shift towards evaluating auditory scenes, known as Soundscapes. Ismail [32] argued that soundscape evaluations consider not only the physical aspects of sound but also the subjective human perceptions. This shift has spurred growth in soundscape research, emphasizing the need for sophisticated management of sonic environments [33]. Soundscape evaluation methodologies typically include in situ surveys by observers to capture auditory perceptions at specific locations [34,35], and the use of audio recordings from handheld recorders and binaural microphones for off-site response assessments [36,37]. Innovative studies are employing technologies like head-mounted displays and virtual reality to explore urban environments, focusing on aesthetic preferences [38] and overall satisfaction [39], though such methods have limited scope in urban settings [40], which may impact the generalizability of results. The challenge remains in developing automated methods for extracting sound sources from recordings within the dynamic, unstructured urban contexts, which hampers the scalability of soundscape studies over larger areas [41]. Nonetheless, understanding the diverse sound sources in urban areas, such as sidewalks and pedestrian routes, is crucial for creating context-aware autonomous systems. These systems enhance safety in dynamic pedestrian environments by leveraging rich auditory data, thus addressing current technological limitations.

To enhance our understanding of the acoustic environment within the DELTA dataset, particularly in pedestrian pathways, we integrated automated audio event classification by employing deep learning models. This initiative aimed to categorize the diverse soundscapes recorded. Specifically, we utilized two prominent deep learning models for this task: YAMNet [42] and YOLOv8 [43]. YAMNet, pre-trained on Google’s comprehensive AudioSet [44], provides a solid analytical foundation for our study. The extensive range and meticulous categorization of audio events in AudioSet were instrumental in preparing YAMNet to accurately identify a wide array of sounds. This capability aligns perfectly with the objectives of our research, allowing us to deepen our insights into the acoustic landscape captured in our dataset. Utilizing the YAMNet architecture, which integrates the MobileNet-v1 [45] design known for its depthwise-separable convolutions, our study delved into classifying audio event classes from a curated collection of audio segments. This began with normalizing the amplitude range of the audio files, creating a uniform base for analysis. We aligned YAMNet’s parameters, such as sampling rate and frame hop time, with our audio signals’ characteristics to ensure precise temporal representation. To comply with YAMNet’s input standards, stereo audio files were converted into mono format by averaging the channels. Following this preprocessing, each audio segment underwent analysis via YAMNet, which produced detailed spectrograms and class activation maps for each segment. The spectrograms shed light on the frequency distribution and energy patterns within the audio, whereas the class activation maps provided probabilistic evaluations of various audio event categories. An example of YAMNet’s output, as applied to a representative audio segment, is depicted in Figure 8. In conjunction with our use of YAMNet for audio event classification, we adapted the YOLOv8 model to operate within the auditory domain, leveraging its well-known capabilities in visual object detection. Our approach involved segmenting the audio data into 5 s intervals and transforming them into spectrograms for subsequent analysis. For training and validation, we employed the ESC-50 dataset, which provides a diverse range of audio recordings spanning 50 distinct categories. However, to align our model’s focus with the specific objectives of our environmental audio classification study, we selectively trained and validated only 13 classes from the ESC-50 dataset that were most relevant to our work. Table 2 presents a selection of class events identifiable by the models, derived from two datasets that were used in our study.

We adapted the work by [46] for transforming audio signals into their visual representations, specifically spectrograms, to facilitate the classification of environmental sounds. The methodology involves several computational steps and mathematical transformations, as detailed below:**Fourier Transformation:** The audio signal is processed through a Fourier transformation, decomposing a function of time (the audio signal) into its constituent frequencies. The Fourier transform F(ω) of a signal x(t) is given by the following:
$$F\left(\omega \right)=\int_{-\infty }^{\infty }x\left(t\right)e^{-j\omega t}\;dt$$We apply a window function to the signal before performing the Fourier transform to mitigate spectral leakage.**Windowing and Frame Segmentation:** The audio signal is segmented into overlapping frames, and a Hanning window is applied to each frame. The Hanning window w[n] is defined as:
$$w\left[n\right]=0.5-0.5\cos\left(\frac{2\pi n}{N-1}\right)$$
where *N* is the window length. This windowing smooths the signal within each frame, reducing boundary discontinuities.**Spectrogram Creation and Logarithmic Scaling:** The spectrogram is generated by computing the Fast Fourier Transform (FFT) of each windowed frame then applying logarithmic scaling to the frequency axis. The spectrogram S(t,f) can be represented as follows:
$$S\left(t,f\right)={\left|\mathrm{FFT}\{x(t)\cdot w(t)\}\right|}^{2}$$Logarithmic scaling emphasizes lower frequencies, aligning the representation closer to human auditory perception.**Decibel Conversion:** The magnitude of the FFT is converted to decibels (dB), a logarithmic unit expressing the ratio of a physical quantity relative to a specified reference level. The conversion to decibels is given by the following:
$$dB=20\log_{10}\left(\frac{\left|S(t,f)\right|}{\mathrm{ref}}\right)$$
where ref is the reference power level.**Visual Representation:** The resulting spectrogram is converted to an image using a color map. This visual representation facilitates an intuitive understanding of the sound’s characteristics, including frequency content and temporal evolution.

Through this approach, we have trained a YOLOv8 model to identify different sounds based on their visual representation. The training trajectory showed a consistent reduction in loss and a corresponding increase in accuracy, culminating in a top-one accuracy of 69.2% and a top-five accuracy of 92.3% as depicted in Figure 9a. Further, Figure 9b,c visually compare actual and predicted sound classifications through spectrograms, highlighting the model’s ability in discerning sound signatures from diverse sources.

## 6. Sidewalks and Pedestrian Route Segmentation

Sidewalks and pedestrian routes, defined as designated or commonly used pathways for foot traffic, are fundamental to the integration and success of autonomous delivery robotics and micro-mobility services, where ensuring safety and efficient navigation is paramount. The accurate segmentation of these areas is crucial for enabling delivery robots and micro-mobility vehicles to safely coexist with pedestrians by adhering to designated pathways and minimizing the risks of accidents. This segmentation not only improves route planning but also enhances context-awareness across diverse urban environments. In this research, we utilized the SegFormer architecture [47] for semantic segmentation of sidewalks and pedestrian pathways, we analyzed three distinct datasets, each comprising 950 images, for a total of 2850 images. These datasets included imagery from the right image sensor of ZED2i, Osmo 3 camera footage, and reflectivity data from an Ouster LiDAR. This channel captures the varying intensities of laser light reflected back from different materials, which provides a natural and reliable contrast, making it highly effective for identifying buildings, vegetation, and roads, especially in scenarios with fluctuating lighting conditions. Unlike traditional geometric data from LiDAR, which focuses on the 3D structure of the environment, the reflectivity channel delves deeper into material properties. By incorporating both LiDAR’s geometric data and the reflectivity channel it is possible to gain more accurate understanding of the environment. Simultaneously, the high resolution of the 4K imagery from the Osmo 3 camera contributes significantly by offering higher pixel density of standard 1080p HD images. This increased resolution allows for more detailed distinctions within the image, crucial for the accurate segmentation and identification of intricate features. Conversely, the imagery from the ZED2i sensor, though lower in resolution, is crucial for its efficiency and processing speed, particularly suitable for real-time applications. It offers a broader context or general shapes, simplifying the segmentation process and reducing computational load and storage requirements, which is advantageous for mobile platforms where quick assessments are necessary. Collectively, these varied resolution types play complementary roles, enhancing the scalability, accessibility, and precision of our semantic segmentation efforts.

In our work, we utilize a transfer learning approach for semantic segmentation, employing the pre-trained variant of the SegFormer model. This model serves as a robust feature extractor, specifically configured to adapt to our segmentation needs by disabling its internal label reduction mechanism to handle more than the standard 512 classes and resizing input images to (128 × 128) pixels to enhance compatibility and efficiency. Each dataset was manually annotated using the Segment Anything Model (SAM) [48], ensuring precise labeling for accurate segmentation. Following annotation, we optimized data processing using data loaders configured for batches of eight images, supported by two worker threads to enhance throughput. This setup reduced load times and maximized computational efficiency, speeding up the training, validation, and testing phases. The pre-trained model is extended with a final classification layer tailored to our specific class count. This setup includes regular metric evaluations and an early stopping mechanism to halt training if validation loss does not improve, thereby preventing overfitting. Figure 10 displays examples of our custom segmentation models for sidewalks and pedestrian routes, with results highlighted in red and overlaid on the original images for each sensor type.

Table 3 clearly captures the performance metrics for models applied to the reflectivity channel, 4K, and ZED2i datasets. The reflectivity channel model demonstrated good prediction accuracy with a Dice coefficient of 0.8452 and a high classification precision, evidenced by a frequency weighted accuracy of 0.9829 and a mean IoU of 0.8461. The 4K dataset model excelled, with the highest dice coefficient of 0.9676 and an impressive mean IoU of 0.9603, alongside a frequency weighted accuracy of 0.9847, indicating promising segment prediction and classification accuracy. The ZED2i dataset model also showed strong performance with a dice coefficient of 0.9384, a mean IoU of 0.9267, and a frequency-weighted accuracy of 0.9774, showcasing robust segmentation capabilities.

## 7. Use Cases

This section outlines the practical applications of our research, demonstrating both current implementations and future potential uses of the DELTA dataset. Our focus spans across diverse aspects of image-based localization, enriched with multi-contextual georeferenced data, audio event classification, diverse sidewalk segmentation, and detailed 3D reconstruction of urban environments, illustrating the capabilities of the dataset.

### 7.1. Current Implementations Using the DELTA Dataset

**Refining Image-Based Localization Through Tailored Landmark Segmentation Models:** The importance of landmark segmentation extends beyond mere identification; it significantly enhances image-based localization [49] by providing accurate, context-rich cues that improve the precision and reliability of navigation and localization algorithms. To further refine image-based localization systems, we utilized the high-quality visuals and precise geolocations from the DELTA dataset to develop a custom landmark segmentation model. This model specializes in identifying and segmenting key urban landmarks such as historic sites, unique architecture, and public spaces, which are crucial for navigation in complex urban settings.

We annotated 27 key landmarks in approximately 800 street-level images using the SAM and trained an instant segmentation model based on the YOLOv8 architecture [43]. This training was conducted over 100 epochs using a Stochastic Gradient Descent (SGD) optimizer without relying on pre-trained models to ensure the model was tailored specifically to our dataset. An early stopping mechanism was included at 50 epochs to prevent overfitting. The model not only recognizes but also provides contextual details about these landmarks, enhancing the precision of localization algorithms. Equipped with detailed knowledge of an area’s unique visual markers, autonomous navigation systems can achieve better orientation and accuracy, which are crucial for the safe navigation of autonomous vehicles and devices in urban environments. The effectiveness of our landmark segmentation model is visually represented in Figure 11.

**Multi-Contextual Georeferenced Dataset for Image-based Localization Retrieval:** For the task of image-based localization retrieval, we developed a multi-contextual georeferenced dataset using the DELTA dataset and the landmark segmentation model. To construct this map, the landmark segmentation model was applied to street-level imagery to identify and segment key urban landmarks. Each landmark was then correlated with its precise geographic location from the DELTA dataset’s GNSS coordinates, ensuring accurate mapping to real-world geographical counterparts. We then systematically organized this enriched data by downsampling aerial view images and arranging them as tiles in a sequence based on their path trajectory and timestamps. Each tile contains detailed information including the identified landmarks, their specific geolocations, and a positional index. By overlaying these tiles with segmented landmark data and precise geolocation coordinates, we created a detailed, information-rich map. This map visually represents the urban layout and embeds each image with multi-layered contextual data, enhancing the utility for urban navigation and planning. Figure 12 showcases the multi-contextual georeferenced dataset.

### 7.2. Future Practical Use Cases

**Enhancing Urban Autonomous Navigation through Audio Event Classification:** Traditional autonomous navigation systems have predominantly relied on visual data, which, despite their effectiveness, encounter limitations within the multifaceted auditory landscape of urban environments. The integration of audio event classification into these systems can substantially improve their functionality by identifying and interpreting various urban sounds, such as emergency sirens, honking, and the presence of cyclists, thereby enhancing situational awareness and decision-making capabilities. Furthermore, the integration of geotagged audio data from the DELTA dataset combined with event-based audio classification introduces a novel dimension to urban analysis. This combination allows for the mapping of specific city regions with their unique acoustic signatures, providing an additional layer of context. For autonomous systems, this means an enriched understanding of their surroundings beyond visual cues alone, allowing for more nuanced responses to urban challenges. For urban planners and public safety officials, this technology offers a deeper insight into the acoustic dynamics of different city areas, aiding in the development of strategies to enhance pedestrian safety and manage noise pollution more effectively.

**Enhancing Autonomous Navigation with Diverse Sidewalk Segmentation:** Autonomous navigation systems have traditionally relied on generic models for sidewalk detection [50], which may not accurately reflect the diverse characteristics of different urban settings. By integrating customized pathway segmentation with the multi-resolution and diverse sidewalk and pedestrian route detection capabilities of the DELTA dataset, these systems can achieve a much deeper understanding of urban landscapes. This approach allows for the differentiation not only of paths but also of the context surrounding them, such as clearly distinguishing pedestrian areas from vehicle lanes. Utilizing the DELTA dataset enhances these models by providing varied resolution data that captures a broader spectrum of urban details, enabling real-time adaptation to the dynamic urban environment. This adaptation enhances the safety and efficiency of interactions among autonomous vehicles, pedestrians, and cyclists, making navigation systems more responsive and reliable in complex cityscapes.

**Three-Dimensional Reconstruction of Urban Environments:** Digital representation of urban settings is crucial for effective city planning, simulation, and technological development [51]. It provides a foundational framework that allows urban planners, developers, and researchers to visualize, analyze, and interact with the complex dynamics of city landscapes. Leveraging the DELTA dataset, which offers extensive high-resolution imagery and precise geolocation data, enhances the creation of detailed 3D models of these landscapes. These models are useful for various applications, including urban planning, virtual reality simulations, and the development of autonomous systems. They enable stakeholders to accurately visualize and evaluate proposed changes, offer immersive experiences for detailed exploration, and serve as realistic environments for testing navigation technologies in autonomous vehicles and drones.

## 8. Limitations and Future Scope of Research

In this section, we critically reflect upon the inherent challenges and limitations encountered in the creation and utilization of the DELTA dataset, along with the strategies adopted to address these issues and enhance the dataset’s applicability and robustness for future research and applications in urban mobility. We acknowledge that the DELTA dataset’s initial geographic scope may not fully capture the vast diversity of urban environments, potentially impacting its generalizability across varied urban landscapes and the dynamic interactions within them. Furthermore, the environmental conditions during data collection were not entirely representative of the diverse scenarios encountered in urban settings, such as fluctuating weather conditions. These limitations are crucial for the dataset’s performance and reliability, given the significant impact of environmental factors on sensor-based data acquisition. To mitigate these concerns, future endeavors will aim at broadening the geographic and environmental scope of the dataset. This includes plans to extend data collection efforts beyond the cold season in Estonia, which had initially restricted our ability to capture a wide array of urban settings and pedestrian behaviors. Incorporating data from various seasonal contexts and extending collection efforts to include different times of day and days of the week are pivotal steps toward ensuring that the dataset reflects the variability in environmental conditions and pedestrian activities throughout the year.

By diversifying the temporal and seasonal coverage, we aim to craft a dataset that offers a holistic view of urban pedestrian environments. This effort will significantly bolster the dataset’s diversity, scalability, and applicability for large-scale research and testing across a broad spectrum of urban contexts worldwide. Such enhancements are vital for developing technologies that are adaptable to the complex dynamics of urban environments, thereby supporting the advancement of urban planning, autonomous navigation, and the development of pedestrian-friendly urban infrastructure.

Our aspiration is for the DELTA dataset to act as a catalyst for further research and methodological innovation in the study of urban environments. We understand the critical need for precise metrics to gauge the dataset’s effectiveness for particular research initiatives. Therefore, we aim to foster a collaborative environment with the research community, working together to establish these essential benchmarks.

Moving forward, our strategy to enhance the comprehensiveness and applicability of the DELTA dataset involves two key initiatives: Firstly, we plan to augment our data collection efforts by incorporating a fleet of e-scooters. This decision stems from the success of using an e-scooter platform, which has proven exceptionally agile and efficient in navigating a variety of urban environments. The e-scooter’s ability to access narrow pathways, pedestrian zones, and cycling tracks enables us to capture a more diverse array of urban settings, enriching the dataset with a comprehensive view of urban pedestrian environments. Secondly, to ensure the dataset remains current and reflective of the ever-changing urban landscapes, we will implement a schedule of regular data collection sessions. The flexibility and rapid deployment capabilities of the e-scooter platform make it an ideal tool for frequent updates, allowing us to maintain the dataset’s relevance and accuracy over time. By expanding to a fleet of e-scooters for data collection, we anticipate significant increases in the volume and diversity of data captured.

Our commitment to these future directions is aimed at continuously improving the DELTA dataset, ensuring that it becomes an even more valuable resource for urban studies, particularly in the domains of pedestrian behavior analysis, autonomous navigation, and urban planning.

## 9. Conclusions

In this study, we presented the DELTA dataset, a meticulously assembled multimodal dataset specifically designed to deepen the understanding and analysis of pedestrian routes within diverse urban environments. Utilizing a custom-designed, multi-sensor e-scooter platform, we gathered high-resolution and synchronized data across audio, visual, LiDAR, and GNSS/IMU sensors. This robust integration furnishes a detailed and contextually rich view of urban pedestrian dynamics, capturing the intricate interactions within these spaces.

We developed three innovative pedestrian route segmentation models, each optimized to capitalize on the distinct characteristics of the corresponding sensor type. These models demonstrated high efficacy in accurately delineating pedestrian pathways across a variety of urban landscapes. Furthermore, by employing a pre-trained YAMNet model and developing our own audio event classifier using the YOLOv8 architecture, we have enhanced our dataset by associating distinct auditory landscapes with their precise geographic locations, thereby adding layers of contextual urban data.

The DELTA dataset holds significant potential for applications such as enhancing autonomous navigation systems and advancing urban planning, yet its influence extends further. As urban environments evolve, the insights from the DELTA dataset support the development of urban mobility solutions that are efficient, inclusive, and tailored to meet the diverse needs of city dwellers. We are planning to expand the dataset to cover a broader spectrum of environments and conditions, which will modestly improve its precision and relevance for urban studies and smart city developments. This gradual expansion aims to make the dataset a useful resource for understanding and addressing urban challenges.

## Figures and Tables

**Figure 1 sensors-24-03863-f001:**
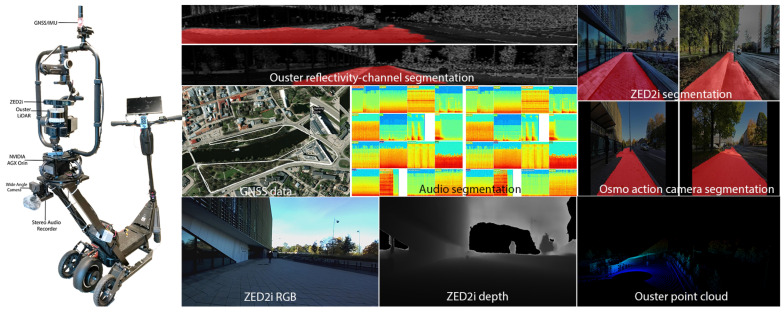
An overview of the platform, showcasing our multisensor datasets, segmentation techniques, sound event classifications, and the study area in Tartu, Estonia.

**Figure 2 sensors-24-03863-f002:**
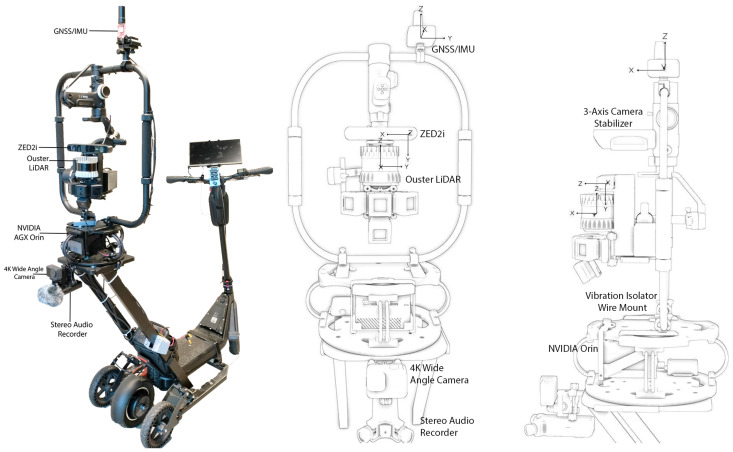
Our multi-sensory mobile capturing platform followed by the coordinate system for each sensor setup.

**Figure 3 sensors-24-03863-f003:**
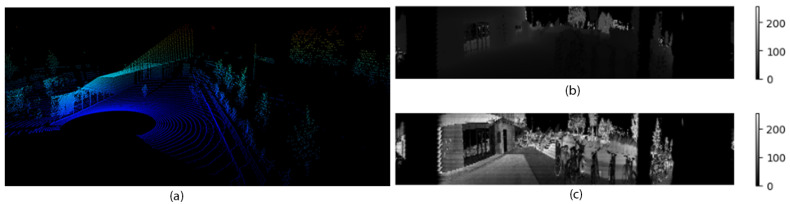
Data formats found in our LiDAR dataset: (**a**) point cloud, (**b**) range, and (**c**) reflectivity channel.

**Figure 4 sensors-24-03863-f004:**
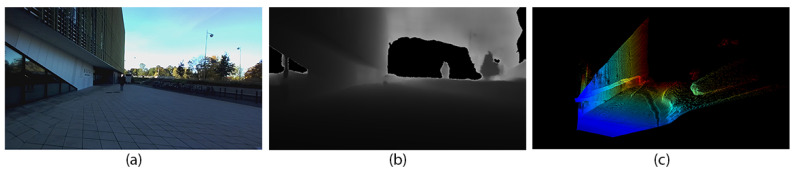
Data formats found in our stereocamera dataset: (**a**) RGB, (**b**) depth map, and (**c**) point cloud.

**Figure 5 sensors-24-03863-f005:**
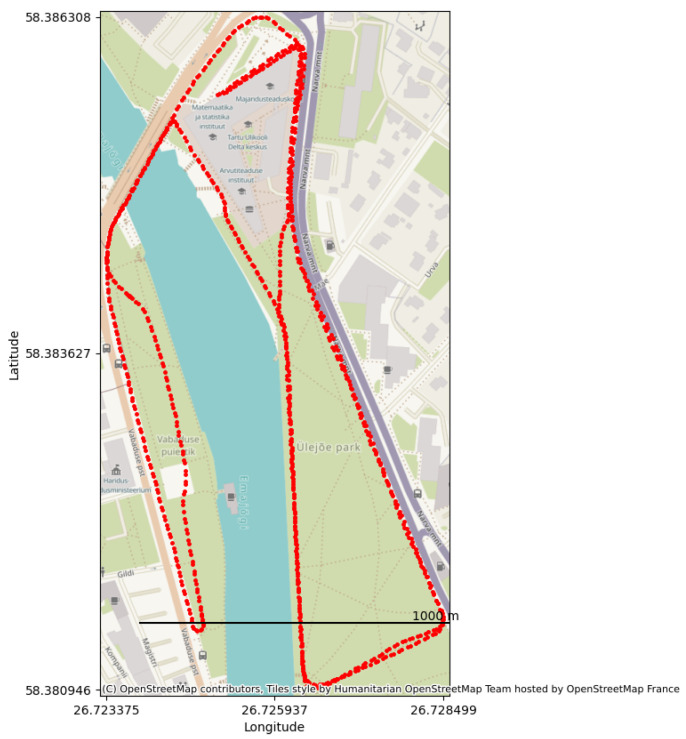
Georegistered trajectory points mapped on OpenStreetMap.

**Figure 6 sensors-24-03863-f006:**
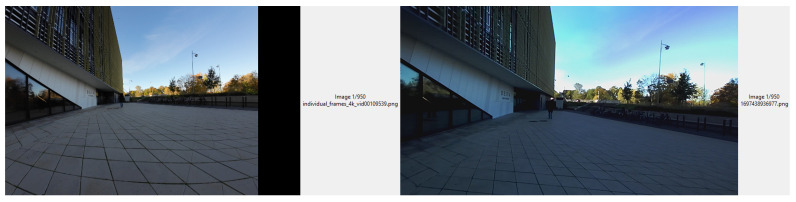
GUI interactive interface for comparative image selection: 4K dataset vs. stereocamera dataset.

**Figure 7 sensors-24-03863-f007:**
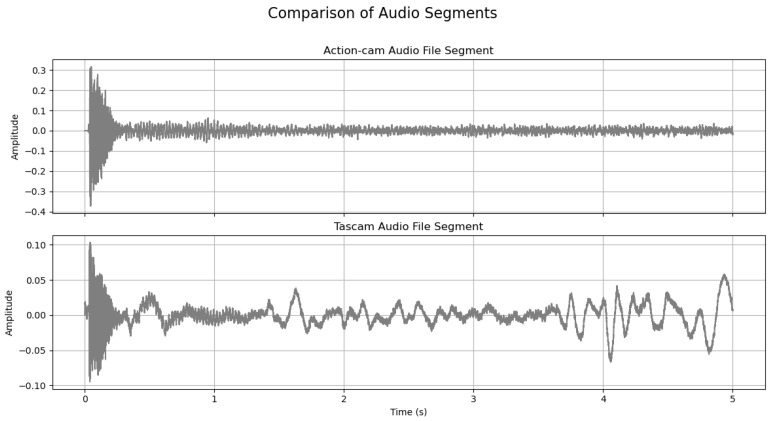
Aligned audio waveforms from an action camera (**top**) and a Tascam recorder (**bottom**), demonstrating synchronization over a 5 s interval with corresponding time (s) and amplitude variations.

**Figure 8 sensors-24-03863-f008:**
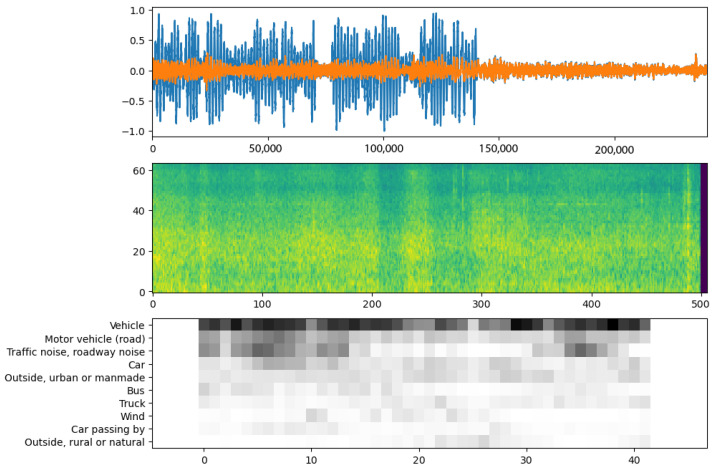
Audio analysis visualizations generated using YAMNet. The top panel shows the normalized stereo waveform of the audio chunk. The middle panel presents the corresponding spectrogram, indicating frequency content over time with color intensity representing energy levels. The bottom panel illustrates the class activation map, displaying the probabilities of various audio event classes detected across the audio timeline.

**Figure 9 sensors-24-03863-f009:**
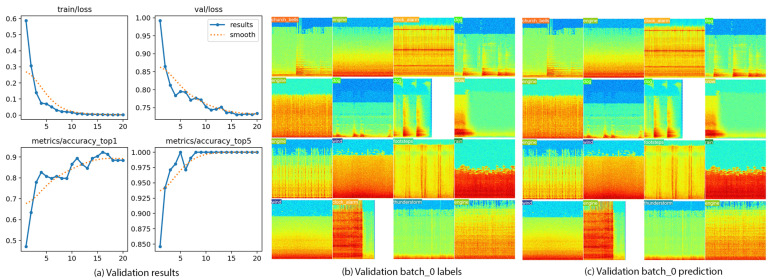
(**a**) Validation results from YOLOv8 audio classification model, (**b**) spectrograms for validation audio samples with true labels versus (**c**) model predictions, depicting sound signatures for various sources.

**Figure 10 sensors-24-03863-f010:**
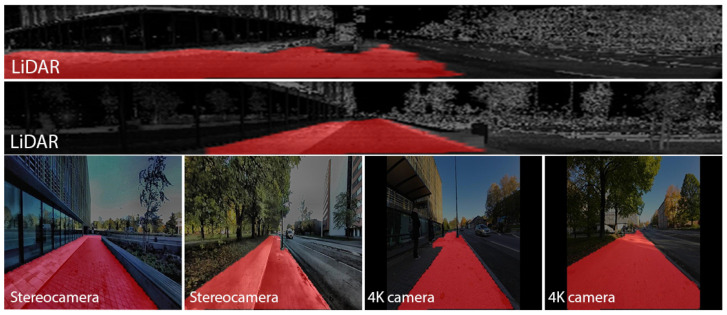
Examples of sidewalk and pedestrian route segmentation: Displaying six segmented images from LiDAR (reflectivity channel), stereocamera (right-side image), and monocular camera with overlaid color-coded label (highlighted in red) from various geographical regions.

**Figure 11 sensors-24-03863-f011:**
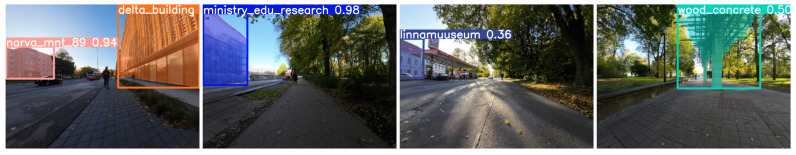
Examples of our custom landmark segmentation in street-level images captured in Tartu, Estonia.

**Figure 12 sensors-24-03863-f012:**
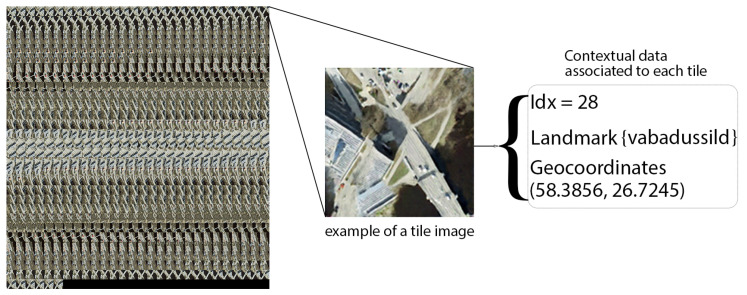
Graphic representation for our multi-contextual georeferenced dataset.

**Table 1 sensors-24-03863-t001:** Sensor specifications of our platform.

Sensor (Manufacturer)	Details
Stereocamera (Stereolabs, Paris, France)	Dual 1/3″ 4MP CMOS
	FOV: 110° (H) × 70° (V) × 120° (D)
	Range: 0.3 m to 20 m
	Gyroscope, Accelerometer
	Magnetometer, Barometer, Temperature.
	Resolution: (720 × 405) 32-Bit.
	Frame Rate = 58 FPS
LiDAR (Ouster, San Francisco, CA, USA)	Resolution: (1024 × 64) 32-Bit
	FOV: 360°
	Range: 200 m
	Sampling frequency: 10 Hz
Action Camera (DJI, Shenzhen, China)	4K (4096 × 3072)
	Frame Rate = 30 FPS
	FOV: 155°
	Audio Sampling Frequency: AAC 48.0 kHz
Audio Recorder (Tascam, Tokyo, Japan)	Built-in cardioid condenser stereo pair
	Sampling frequency: 96 kHz/24-Bit
IMU (WitMotion, Shenzhen, China)	Acceleration, Gyroscope, Magnetometer
	Angle (X, Z-axis: ±180°, Y ±90°)
	Sampling frequency: 100 Hz
GNSS module (u-blox, Thalwil, Switzerland)	Receives both L1C/A and L2C bands
	PVT (basic location over UBX binary protocol)
	Horizontal accuracy of 1.11 m
	Sampling frequency: 2 Hz

**Table 2 sensors-24-03863-t002:** Comparison of 5 classes from Audioset and ESC-50 datasets.

Audioset Classes	ESC-50 Classes
vehicle	siren
traffic noise	rain
roadway noise	car horn
car	engine
truck	footsteps

**Table 3 sensors-24-03863-t003:** Results of segmentation model evaluation on three datasets.

Dataset	Dice Coefficient	Frequency Weighted Accuracy	Mean IoU
Reflectivity channel	0.8452	0.9829	0.8461
4K	0.9676	0.9847	0.9603
ZED2i	0.9384	0.9774	0.9267

## Data Availability

The entire dataset (both sensory data and models) is available at https://its.cs.ut.ee/home/resources (accessed on 9 June 2024).

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
