# Peer review of "DELTA: Integrating Multimodal Sensing with Micromobility for Enhanced Sidewalk and Pedestrian Route Understanding"

_sensors, 2024, doi:10.3390/s24123863_

Round 1

Reviewer 1 Report

Comments and Suggestions for Authors

In Section 5, Data Synchronization, the paper describes the techniques and methods of sensor data synchronization, but does not provide detailed information on how to process the synchronized data to ensure data quality.

Although the paper introduces the pedestrian route segmentation model, it lacks a comparison with prior art and the details of performance evaluation are insufficient.

Author Response

Thank you for your insightful review of our manuscript. Your feedback has been incredibly helpful in improving our work. We truly appreciate your time and expertise.

Reviewer 2 Report

Comments and Suggestions for Authors

The paper addresses the evolving nature of pedestrian areas into complex transportation hubs and the limitations of road-centric datasets in capturing pedestrian spatial details. It introduces the DELTA dataset, focusing on pedestrian pathways to meet the demands of pedestrian spatial analysis. Different pedestrian route segmentation models are developed for various sensors to ensure accurate route recognition. Additionally, audio event-based classification enriches the understanding of urban environments. The paper discusses the application and limitations of the DELTA dataset. However, it's important to note that the article's overall structure and the presentation of results lack clarity and logical flow.

Comments and suggestions are as follows:

1.       In the abstract, it is vital to outline the effectiveness of different models in pedestrian route segmentation and the results of audio event classification.

2.       The article's language requires refinement. It appears repetitive and redundant at times and needs streamlining, such as 3, 4.4.

3.       The second paragraph of the introduction is redundant. Please divide it into multiple paragraphs, with particular emphasis on highlighting the contributions of this article.

4.       Figure 4 (b) and (c) have titles, while Figure 4(a) does not. Unify the style.

5.       In section 5, it is unnecessary to include the statement, "This removes the need for external timing sources or complex synchronization methods, ensuring all sensor data are precisely timed, maintaining consistent temporal alignment, and boosting project efficiency." The description in the article is overly redundant. It is better to state the adopted strategies and methods directly.

6.       The core methods of the article are presented in section 6 and section 7. However, the detailed process of establishing the models is lacking, with no emphasis on the key points or highlighting the work presented in this article.

7.       In section 8, the description of the practical application of the DELTA dataset is enhanced by providing additional supporting evidence rather than plain textual descriptions.

Comments on the Quality of English Language

The article's language should be polished.

Author Response

(The authors gave the same response as above.)

Reviewer 3 Report

Comments and Suggestions for Authors

DELTA: Integrating Multimodal Sensing with Micromobility for Enhanced Sidewalk and Pedestrian Route Understanding

In this manuscript, the authors address the limitations of traditional, road-centric datasets in capturing pedestrian space dynamics by introducing the DELTA dataset, designed to enhance the analysis and mapping of pedestrian zones. They collected this dataset from a single urban setting using a custom multi-sensing e-scooter platform, gathering high-resolution and synchronized audio, visual, LiDAR, and GNSS/IMU data. They developed three pedestrian route segmentation models tailored to the 4K camera, stereocamera, and LiDAR, leveraging each sensor's strengths for accurate route identification. Additionally, they explored audio event-based classification to link unique soundscapes with specific geolocations, enriching the spatial understanding of urban environments. They conclude by discussing potential use cases, limitations, and future research directions for better understanding pedestrian environments.

This manuscript is very interesting, and the authors have made a significant effort in presenting the work and reviewing previous research, incorporating methodologies from various studies. After a comprehensive review, I have observed some points that the authors should properly address to make this work acceptable. Here are my concerns and suggestions:

·         First, better present the complexity of urban spaces from a global context. Cities are very different across regions, and our mobility models are influenced by them. We could say that cities modify our mobility models, among many other aspects. Some recent works align with this perspective, such as the recent work "How cities influence social behavior" in Digital Ethology (2024): From Individuals to Communities and Back. Strüngmann Forum Report, vol. 34, edited by Tomáš Paus and Hye-Chung Kum. Cambridge, MA, MIT Press.

·         The authors state in the introduction, “In this urban evolution, the various aspects of pedestrian infrastructure play a pivotal role [1], sharing space with these emerging modes of urban mobility.” In urbanism, concepts such as conflict zones (between mobility models) and shared space are often discussed.

·         Right after, they argue, “Pedestrian elements, such as sidewalks, crosswalks, pedestrian routes, and public pathways, are not mere footpaths but critical components of the modern urban environment, where human mobility harmoniously intersects with the advancement of autonomous and mobile robotic systems [2].” It would be interesting for the authors to discuss the relationship between transportation and urban planning, as both are intertwined. Of course, city mobility depends on the prevailing technology at any given time, which changes not only movement patterns but also how we work. Therefore, addressing this complexity would be beneficial.

·         They also state, “these commonly overlooked urban spaces, primarily designed for pedestrian use, are no longer exclusive domains of human foot traffic.” Do the authors think they were ever solely for that? For example, American cities were designed for car mobility, while people moved to the suburbs.

·         “Instead, they are becoming dynamic environments where diverse modes of mobility coexist. Electric scooters, e-bikes, autonomous delivery. These transformations raise questions about safety, efficiency, and accessibility in shared urban spaces, necessitating a nuanced and data-driven approach to urban robots, and future technologies like autonomous personal mobility devices are increasingly becoming part of the urban fabric [3].” This is an important point. It is not only about futuristic methods. For example, the fact that electric cars are quieter already signifies a major change.

·         When the authors introduce their dataset, they mention a large number of technologies, “The sensor diversity within our platform ranges from a high-resolution 4K monocular camera, stereo camera, and LiDAR, to audio recording devices, IMU, and GNSS receiver, ensuring a rich, multidimensional dataset.” Here, an important point arises. Are the data captured simultaneously with a single vehicle or in several passes? How are the data integrated if captured at different frequencies? This is a significant issue in such experiments.

·         The authors argue that “The DELTA dataset includes sequences of stereo RGB images, depth maps, and point clouds generated from disparity maps for detailed visual representations, alongside sequences of 3D point clouds, reflectivity, and range channel data from LiDAR for precise spatial measurements.” How do they handle areas where certain technologies present low resolution or data? For example, LiDAR can present very low densities in certain coverages and depending on the optical mirror used. For instance, see Lerma et al. (2014) on the Empirical study of variation in lidar point density over different land covers. I recommend the authors evaluate this aspect and include more references to relevant works.

·         I suggest the authors explain more clearly the cleaning and preparation of the database from the raw data.

·         From line 28 to 77, it is all one paragraph. The authors need to structure it into smaller, clearer paragraphs, which are easier to read.

·         The authors state, “Pseudorange data offers an initial estimate of the distance between the satellite and the receiver, while the carrier phase data affords a more refined measurement critical for differential GNSS applications that require heightened accuracy.” At times, it is difficult to follow this work due to the large number of technologies and concepts introduced, which I believe could be simplified (Pseudorange is common knowledge). My question is whether the authors can simplify the presentation so that a general non-expert reader can understand the most relevant aspects.

·         Just few lines before, the authors argue “It includes pseudorange, carrier phase, and Doppler shift data from multiple satellite constellations, namely GPS, GLONASS, Galileo, and BeiDou.” (a) This is generally known as GNSSs, global system of GNSS, or more simply, interoperability between GNSS. (b) I think the Chinese GNSS is now called Compass, not Beidou, which was the regional system.

·         The authors state this about data integration, “Moreover, the synchronization of this IMU data with precise timestamps and GNSS coordinates significantly improves the accuracy of location data. The application of advanced data fusion techniques – including the Kalman filter[36], Extended Kalman Filter[37], and Fuzzy Logic Systems[38] – in conjunction with accelerometer data, further refines the quality and functionality of the mapping system.” An interesting work I can suggest, though mostly in Spanish, is the PhD dissertation titled the Application of GNSS and GIS systems on transport infrastructures, where the author works on many of the points presented here, such as integrating information from various sensors in a vehicle, including GNSS, IMU, and others. The dissertation body is in Spanish but dedicates a chapter to this integration and discusses the general problem of synchronizing technologies during data collection. I recommend expanding this point by including more relevant references like the one mentioned.

·         In the Figure 6. “Georegistered trajectory points overlaid on OpenStreetMap” the authors should include a spatial scale.

·         Summarize the conclusions, presenting the results achieved in this manuscript more succinctly and clearly.

·         In Figure 7, “GUI interface for manually selecting similar images from two datasets” -> It is not very clear what this figure is meant to represent, especially the text on the right side.

·         Finally, I observed unclear use of certain expressions, the wrong use of past forms and writing errors. For example, the use of capitalization in line 72, where the authors say, “Third, We developed…”.

Comments on the Quality of English Language

DELTA: Integrating Multimodal Sensing with Micromobility for Enhanced Sidewalk and Pedestrian Route Understanding

In this manuscript, the authors address the limitations of traditional, road-centric datasets in capturing pedestrian space dynamics by introducing the DELTA dataset, designed to enhance the analysis and mapping of pedestrian zones. They collected this dataset from a single urban setting using a custom multi-sensing e-scooter platform, gathering high-resolution and synchronized audio, visual, LiDAR, and GNSS/IMU data. They developed three pedestrian route segmentation models tailored to the 4K camera, stereocamera, and LiDAR, leveraging each sensor's strengths for accurate route identification. Additionally, they explored audio event-based classification to link unique soundscapes with specific geolocations, enriching the spatial understanding of urban environments. They conclude by discussing potential use cases, limitations, and future research directions for better understanding pedestrian environments.

This manuscript is very interesting, and the authors have made a significant effort in presenting the work and reviewing previous research, incorporating methodologies from various studies. After a comprehensive review, I have observed some points that the authors should properly address to make this work acceptable. Here are my concerns and suggestions:

·         First, better present the complexity of urban spaces from a global context. Cities are very different across regions, and our mobility models are influenced by them. We could say that cities modify our mobility models, among many other aspects. Some recent works align with this perspective, such as the recent work "How cities influence social behavior" in Digital Ethology (2024): From Individuals to Communities and Back. Strüngmann Forum Report, vol. 34, edited by Tomáš Paus and Hye-Chung Kum. Cambridge, MA, MIT Press.

·         The authors state in the introduction, “In this urban evolution, the various aspects of pedestrian infrastructure play a pivotal role [1], sharing space with these emerging modes of urban mobility.” In urbanism, concepts such as conflict zones (between mobility models) and shared space are often discussed.

·         Right after, they argue, “Pedestrian elements, such as sidewalks, crosswalks, pedestrian routes, and public pathways, are not mere footpaths but critical components of the modern urban environment, where human mobility harmoniously intersects with the advancement of autonomous and mobile robotic systems [2].” It would be interesting for the authors to discuss the relationship between transportation and urban planning, as both are intertwined. Of course, city mobility depends on the prevailing technology at any given time, which changes not only movement patterns but also how we work. Therefore, addressing this complexity would be beneficial.

·         They also state, “these commonly overlooked urban spaces, primarily designed for pedestrian use, are no longer exclusive domains of human foot traffic.” Do the authors think they were ever solely for that? For example, American cities were designed for car mobility, while people moved to the suburbs.

·         “Instead, they are becoming dynamic environments where diverse modes of mobility coexist. Electric scooters, e-bikes, autonomous delivery. These transformations raise questions about safety, efficiency, and accessibility in shared urban spaces, necessitating a nuanced and data-driven approach to urban robots, and future technologies like autonomous personal mobility devices are increasingly becoming part of the urban fabric [3].” This is an important point. It is not only about futuristic methods. For example, the fact that electric cars are quieter already signifies a major change.

·         When the authors introduce their dataset, they mention a large number of technologies, “The sensor diversity within our platform ranges from a high-resolution 4K monocular camera, stereo camera, and LiDAR, to audio recording devices, IMU, and GNSS receiver, ensuring a rich, multidimensional dataset.” Here, an important point arises. Are the data captured simultaneously with a single vehicle or in several passes? How are the data integrated if captured at different frequencies? This is a significant issue in such experiments.

·         The authors argue that “The DELTA dataset includes sequences of stereo RGB images, depth maps, and point clouds generated from disparity maps for detailed visual representations, alongside sequences of 3D point clouds, reflectivity, and range channel data from LiDAR for precise spatial measurements.” How do they handle areas where certain technologies present low resolution or data? For example, LiDAR can present very low densities in certain coverages and depending on the optical mirror used. For instance, see Lerma et al. (2014) on the Empirical study of variation in lidar point density over different land covers. I recommend the authors evaluate this aspect and include more references to relevant works.

·         I suggest the authors explain more clearly the cleaning and preparation of the database from the raw data.

·         From line 28 to 77, it is all one paragraph. The authors need to structure it into smaller, clearer paragraphs, which are easier to read.

·         The authors state, “Pseudorange data offers an initial estimate of the distance between the satellite and the receiver, while the carrier phase data affords a more refined measurement critical for differential GNSS applications that require heightened accuracy.” At times, it is difficult to follow this work due to the large number of technologies and concepts introduced, which I believe could be simplified (Pseudorange is common knowledge). My question is whether the authors can simplify the presentation so that a general non-expert reader can understand the most relevant aspects.

·         Just few lines before, the authors argue “It includes pseudorange, carrier phase, and Doppler shift data from multiple satellite constellations, namely GPS, GLONASS, Galileo, and BeiDou.” (a) This is generally known as GNSSs, global system of GNSS, or more simply, interoperability between GNSS. (b) I think the Chinese GNSS is now called Compass, not Beidou, which was the regional system.

·         The authors state this about data integration, “Moreover, the synchronization of this IMU data with precise timestamps and GNSS coordinates significantly improves the accuracy of location data. The application of advanced data fusion techniques – including the Kalman filter[36], Extended Kalman Filter[37], and Fuzzy Logic Systems[38] – in conjunction with accelerometer data, further refines the quality and functionality of the mapping system.” An interesting work I can suggest, though mostly in Spanish, is the PhD dissertation titled the Application of GNSS and GIS systems on transport infrastructures, where the author works on many of the points presented here, such as integrating information from various sensors in a vehicle, including GNSS, IMU, and others. The dissertation body is in Spanish but dedicates a chapter to this integration and discusses the general problem of synchronizing technologies during data collection. I recommend expanding this point by including more relevant references like the one mentioned.

    Finally, I observed unclear use of certain expressions, the wrong use of past forms and writing errors. For example, the use of capitalization in line 72, where the authors say, “Third, We developed…”.

Author Response

(The authors gave the same response as above.)

Round 2

Reviewer 2 Report

Comments and Suggestions for Authors The suggestions given earlier in the article have been sufficiently corrected. I think it can be published as it is.